# Histories of Dermatan Sulfate Epimerase and Dermatan 4-*O*-Sulfotransferase from Discovery of Their Enzymes and Genes to Musculocontractural Ehlers-Danlos Syndrome

**DOI:** 10.3390/genes14020509

**Published:** 2023-02-16

**Authors:** Shuji Mizumoto, Shuhei Yamada

**Affiliations:** Department of Pathobiochemistry, Faculty of Pharmacy, Meijo University, 150 Yagotoyama, Tempaku-ku, Nagoya 468-8503, Japan

**Keywords:** chondroitin sulfate, dermatan sulfate, dermatan sulfate epimerase, dermatan 4-*O*-sulfotransferase, Ehlers-Danlos syndrome, glycosaminoglycan, proteoglycan

## Abstract

Dermatan sulfate (DS) and its proteoglycans are essential for the assembly of the extracellular matrix and cell signaling. Various transporters and biosynthetic enzymes for nucleotide sugars, glycosyltransferases, epimerase, and sulfotransferases, are involved in the biosynthesis of DS. Among these enzymes, dermatan sulfate epimerase (DSE) and dermatan 4-*O*-sulfotranserase (D4ST) are rate-limiting factors of DS biosynthesis. Pathogenic variants in human genes encoding DSE and D4ST cause the musculocontractural type of Ehlers-Danlos syndrome, characterized by tissue fragility, joint hypermobility, and skin hyperextensibility. DS-deficient mice exhibit perinatal lethality, myopathy-related phenotypes, thoracic kyphosis, vascular abnormalities, and skin fragility. These findings indicate that DS is essential for tissue development as well as homeostasis. This review focuses on the histories of DSE as well as D4ST, and their knockout mice as well as human congenital disorders.

## 1. Introduction

Dermatan sulfate (DS) is classified as a sulfated glycosaminoglycan (GAG) that is a linear polysaccharide chain, and it is covalently attached to the core proteins of proteoglycans (PGs) [1,2,3,4,5]. DS was first isolated from porcine skin by Karl Meyer in the 1940s [6]. DS-PGs have been purified from bovine tendon [7], bovine sclera [8], porcine skin [9], calf skin [10], and bovine periodontal ligament [11]. Two distinct populations of DS-PGs have been found in bovine sclera and cartilage [8,12]. A complete amino acid sequence of a DS-PG, decorin, has been deduced for human [13], bovine [14], chick [15], and rat [16] forms (Table 1). A complete amino acid sequence of another DS-PG, biglycan, has been identified for human [17] and bovine [18] forms. A DS-PG, PG-Lb/epiphycan, has been identified from chick embryonic epiphyseal cartilage [19]. Furthermore, a soluble DS-PG, endocan, has been isolated as an endothelial cell-specific molecule [20,21].

The DS side chain of DS-PGs is composed of an alternating disaccharide unit, *N*-acetyl-D-galactosamineβ1–4L-iduronic acid (-3GalNAcβ1–4IdoUAα1-)_n,_ with 50–200 repeats that are modified by sulfation (Figure 1). The repeating disaccharide region of DS is covalently bound to the serine (Ser) residue of a specific core protein through the common GAG-linker region tetrasaccharide, glucuronic acidβ1-3galactoseβ1-3galactoseβ1-4xyloseβ1-*O*-Ser (GlcUA-Gal-Gal-Xyl-*O*-Ser) [22,23,24,25,26,27] (Figure 1). DS-PGs encompassing DS side chains control a wide range of biological functions including the signal transduction via binding to effector proteins, anti-coagulation, wound healing, and assembly of extracellular matrices [2,3,4,5,28,29,30,31,32,33,34,35]. Thus, DS side chains on PGs are essential for normal development as well as the maintenance of fundamental biological functions. This review focuses on histories of the DS-biosynthetic enzymes dermatan sulfate-epimerase (DSE) and dermatan 4-*O*-sulfotransferase (D4ST).

## 2. Biosynthesis of DS Side Chains on PGs

### 2.1. GAG-Protein Linker Region

Initial biosynthesis of DS side chains is evoked by the construction of a common GAG-protein linker region tetrasaccharide, GlcUA-Gal-Gal-Xy-*O*-Ser (Figure 2) [1,22,23,24,25,26,27]. *XYLT1* and *XYLT2* encode β-xylosyltransferases (XylT), which transfer a Xyl moiety from uridine diphosphate (UDP)-Xyl to specific serine residues in the core proteins of PGs biosynthesized in the endoplasmic reticulum and/or cis-Golgi compartments [36,37,38,39,40,41,42,43,44]. *B4GALT7* encodes β1,4-galactosyltransferase-I (GalT-I), which transfers a β4Gal moiety from UDP-Gal to Xyl-*O*-Ser in the core proteins of PGs in the Golgi apparatus [37,45,46,47,48,49]. GalT-I interacts with XylT, which has been demonstrated through co-immunoprecipitation and the course purification of GalT-I [50,51]. *B3GALT6* encodes β1,3-galactosyltransferase-II (GalT-II), which transfers a β3Gal moiety from UDP-Gal to Galβ1-4Xylβ1-*O*-Ser in the core proteins of PGs [45,46,52]. *B3GAT3* encodes β1,3-glucuronosyltransferase-I (GlcAT-I), which transfers a β3GlcUA moiety from UDP-GlcUA to Galβ1-3Galβ1-4Xylβ1-*O*-Ser in the core proteins of PGs (Figure 2) [53,54,55,56,57].

### 2.2. Repeating Disaccharide Region of DS

After the formation of the linker region tetrasaccharide, GlcUAβ1-3Galβ1-3Galβ1-4Xylβ1-O-Ser, the first GalNAc residue is transferred to the tetrasaccharide-Ser on specific core proteins of PGs by β1,4-*N*-acetylgalactosaminyltransferase-I (GalNAcT-I) (Figure 2) [58,59,60]. *CSGALNACT1* and *CSGALNACT2* genes encoding GalNAcT-I have been identified [61,62,63,64]. CSGALNACT1 and CSGALNACT2 show GalNAcT-II activity (see below) in addition to GalNAcT-I activity [61,62,63,64].

After the attachment of the first β4GalNAc by GalNAcT-I, the resultant nascent pentasaccharide, GalNAcβ1-4GlcUAβ1-3Galβ1-3Galβ1-4Xylβ1-*O*-Ser, is elongated further by alternate additions of β3GlcUA and β4GalNAc from UDP-GlcUA and UDP-GalNAc, which are catalyzed by CS-β1,3glucuronyltransferase-II (CS-GlcAT-II) and β1,4*N*-acetylgalactosaminyltransferase-II (GalNAcT-II), respectively (Figure 2) [65,66,67,68,69,70,71,72]. Chondroitin synthase (ChSy), a single polypeptide chain, has dual enzymatic activities, CS-GlcAT-II and GalNAcT-II [73], whereas chondroitin polymerizing factor (ChPF) showed only weak GalNAcT-II activity [74]. In spite of the dual enzymatic activities of ChSy, ChSy itself cannot achieve polymerization reactions to build up the repeating disaccharide units of chondroitin in vitro. However, the association of ChSy with ChPF, resulting in the enzyme complex, polymerizes a repeating disaccharide region of chondroitin, [-4GlcUAβ1-3GalNAcβ1-]_n_ [74]. To date, four genes, *CHSY1*, *CHSY3*, *CHPF*, and *CHPF2*, have been characterized as CS polymerases (Figure 2) [73,74,75,76,77,78,79].

Following and/or during the formation of the chondroitin precursor chain by ChSy/ChPF family members, the GlcUA moieties are subsequently converted into IdoUA by chondroitin GlcUA-C5-epimerase/dermatan sulfate epimerase (DSE), which epimerizes the C-5 carboxy group of GlcUA residues (Figure 1 and Figure 2) (Table 2) [80,81,82,83,84,85,86,87]. Two genes, *DSE* and *DSEL*, encoding DSE have been identified, and their gene products have demonstrated GlcUA-C5-epimerase activity [88,89,90]. It should be noted that DSE is identical to *SART2* (squamous cell carcinoma antigen recognized by T cells 2), which encodes a protein with unknown function highly expressed in cancer cells and tissues [91], and that *DSEL* is identical to *C18orf4*, which is a candidate gene for a bipolar disorder [92]. 

After and/or during construction of the repeating disaccharide region of dermatan, [-4IdoUAα1-3GalNAcβ1-]_n_, there is further modification with sulfation at the C-2 and C-4 positions of IdoUA and GalNAc residues in dermatan by uronosyl 2-*O*-sulfotransferase (UST) and dermatan 4-*O*-sulfotransferase (D4ST), respectively, using the sulfate donor 3′-phosphoadenosine 5′-phosphosulfate (PAPS) (Table 2) [93,94,95,96,97,98]. *CHST14* and *UST* have been identified as the genes encoding D4ST and UST, respectively [94,95,97]. 4-*O*-Sulfation, but not 2-*O*-sulfation, is predominant in mammals.

## 3. Roles of DSE and D4ST in Biosynthesis of DS

### 3.1. DSE

DSE (also known as chondroitin-glucuronate C5-epimerase) converts GlcUA into IdoUA by epimerization of the C-5 carboxy group of GlcUA moieties in the repeating disaccharide region of the chondroitin precursor chain, [-4GlcUAβ1-3GalNAcβ1-]_n_, to construct the disaccharide region of dermatan, [-4IdoUAα1-3GalNAcβ1-]_n_ (Figure 2). In the 1970s, Malmström et al. demonstrated that the formation of IdoUA from the GlcUA residue occurred at the polymer level together with sulfation using fibroblast extract [80], which suggests the requirement of sulfation for the epimerization of GlcUA. Furthermore, Malmström revealed an important feature of the epimerization of GlcUA into IdoUA: the release of H from C-5 of the target GlcUA [81]. The incorporation of ^3^H into the GlcUA residue of chondroitin from ^3^H_2_O, or the liberated ^3^H, which was released from chondroitin with 5-^3^H-labeled GlcUA residues, can be quantified by liquid scintillation counting [82,83,84]. Dermatan is a better substrate than chondroitin for DSE, from human embryonic fibroblasts, but not CS or DS [83,86]. Tiedemann and Malmström et al. demonstrated high and low levels of chondroitin-glucuronate C5-epimerase activity in human lung fibroblasts and bovine chondrocyte cultures from nasal cartilage, respectively, whereas high 4-*O*-sulfotransferase activity was detected in both cells [87]. These findings suggest that 4-*O*-sulfation on GalNAc residues in both DS and CS may block DSE to fix the configuration of GlcUA and IdoUA.

In 2006, Maccarana et al. identified the gene *SART2*, encoding chondroitin-glucuronate C5-epimerase [88]. The enzyme was purified from the bovine spleen, in which GlcUA C5-epimerase activity is the highest among various tissues examined. Partially purified proteins were subjected to mass spectrometry analysis after treatment with trypsin. SART2, an unknown protein expressed in cancer cells and tissues [91], was identified from the 89-kDa band. There is no orthologous gene in the nematode (*Caenorhabditis elegans*) or fruit fly (*Drosophila melanogaster*). This is consistent with the finding that there is no DS in these organisms [99,100,101]. The recombinant SART2 protein exhibited strong GlcUA C5-epimerase activity. They proposed renaming SART2 as chondroitin-glucuronate C5-epimerase or DS-epimerase [88]. Currently, DSE is widely utilized as the name of the enzyme and gene. Research by Maccarana and Malmström involving the long-term study of DSE has greatly contributed to the elucidation of the functions of DSE as well as DS, and understanding of the pathogenic mechanisms of Ehlers-Danlos syndrome (EDS) with mutations in *DSE*, as described below. 

In 2009, Maccarana et al. generated *Dse*-deficient mice [102], which exhibited reductions in DSE activity as well as DS, a smaller body size, a kinked tail at birth, and thicker collagen fibrils in the dermis and hypodermis [102]. Furthermore, defects in the fetal abdominal wall, exencephaly, spina bifida, and the impaired migration of aortic smooth muscle cells were also detected in *Dse*^–/–^ mice [103,104]. *Dse*-deficient mice also showed a reduction in the migration of skin-derived dendritic cells from the skin to draining lymph nodes that is a prerequisite for encounter with antigens [105]. Consequently, the initiation of the cellular and humoral immune response was diminished in *Dse*^−/−^ mice.

In 2016, Gouignard et al. generated *Xenopus laevis* with knockdown of *Dse* and *Dsel* using morpholino antisense oligonucleotides [106]. Although the knockdown of *Dse* did not affect the formation of early neural crest progenitors, the epimerase activity and extent of neural crest cell migration were decreased in the morphants of *Dse*, which led to a decrease in neural crest-derived craniofacial skeleton, melanocytes, and dorsal fin structures. On the other hand, the knockdown of *Dsel* did not show clear phenotype except for a 12% decrease in DSE activity [106].

DS and/or DS-PGs are elevated in malignant cells and in the stroma [107], which is in agreement with an increase in *DSE* [108]. Thelin et al. demonstrated that a human esophageal squamous cell carcinoma treated with *DSE*-shRNA exhibited reduced migration and invasion abilities in vitro. Furthermore, these cells exhibited reduced cellular interaction with hepatocyte growth factor, inhibition of pERK1/2 signaling, and diminished focal adhesion formation and actin cytoskeleton dynamics [108]. These findings suggest that DS may contribute to the proliferation, invasion, and metastasis of esophageal squamous cell carcinoma via its binding with effector proteins such as hepatocyte growth factor and fibroblast growth factor at the cell surface and/or extracellular matrix.

In 2013, Müller et al. reported that EDS musculocontractural type 2 was caused by the homozygous mutation p.Ser268Leu in *DSE* (Table 2 and Table 3) [109]. The clinical manifestations of their patient presented hypermobility of the finger, elbow, and knee joints, atrophic scars on skin, contracture of the thumbs and feet, and characteristic facial features [109]. DSE activities of recombinant p.Ser268Leu-DSE enzyme, as well as skin fibroblast cultures from the patient, were markedly weaker than in a respective control [109]. It should be noted that Mizumoto et al. developed a methodology to measure DSE activity without using a radioisotope such as 5-^3^H-labeled GlcUA in chondroitin and ^3^H_2_O. Namely, it utilized a CS-specific degrading enzyme, chondroitinase AC, fluorescent labelling, and an anion-exchange HPLC, based on the previous finding that dermatan is a better substrate than chondroitin for DSE [83], and that chondroitinase AC can act on a non-sulfated disaccharide unit, -GlcUA-GalNAc-, but that chondroitinase B, a DS-specific degrading enzyme, cannot act on a non-sulfated disaccharide unit, -IdoUA-GalNAc-. The reaction mixture containing the enzyme source, buffer, and desulfated DS (dermatan), was incubated at 37 °C. The chondroitin moiety of the reaction products with partial conversion from the IdoUA residue into the GlcUA residue can be digested with chondroitinase AC. The resultant di- or oligo-saccharides were labeled with a fluorophore at the reducing end, which was subjected to anion-exchange HPLC. When we analyzed the DSE activity and level of DS in the patient fibroblasts, significant decreases in the amount of DS disaccharide were detected in fibroblasts from the patient, accompanied by a decrease in the DSE activity, compared with those in fibroblasts from a healthy subject [109]. To date, nine cases have been reported as EDS musculocontractural type 2 caused by the mutation, p.Gly216Glufs*3, p.Arg267Gly, p.Ser268Leu, p.Tyr320*, p.Val333Cysfs*4, pPro384Trpfs*9, and pHis588Arg, in *DSE* [109,110,111,112,113]. 

### 3.2. DSEL/DSE2

In 2009, Pacheco et al. demonstrated that the second DSE, DSE2, encoded by *DSEL*, was identified as a homolog of DSE [90]. DSE2 consists of 1222 amino acids, which is larger than DSE (958 aa), has an epimerase domain, ~700 amino acids, at the N-terminus, and a predicted *O*-sulfotransferase domain at the C-terminus [90]. DSE2/DSEL has epimerase activity, but no sulfotransferase activity toward desulfated DS, dermatan, and partially desulfated CS-A and CS-C [90]. The sulfotransferase domain in DSE2 has never been elucidated. It has been proposed as two epimerases, DS-epi1 and DS-epi2, encoded by *DSE* and *DSEL* genes, respectively [90]. To clarify the capacity of DSE and DSE2 in the formation of IdoUA in the DS chain, both genes were overexpressed in human embryonic kidney 293 cells. Overexpression of DSE led to the formation of IdoUA-block in both decorin and biglycan side chains, whereas overexpression of DSE2 resulted in the formation of IdoUA/GlcUA-hybrid structures, but not IdoUA-block [90]. Thus, tissues or cells such as the brain, in which *DSEL* is highly expressed more than *DSE* [114], may produce the CS/DS hybrid chain rather than a genuine DS chain. In fact, CS and DS chains in the mouse brain are predominantly present as hybrid structures containing both GlcUA and IdoUA residues in a single GAG chain [114,115,116]. 

In 2021, Bartolini et al. generated *Dsel*-deficient mice [117]. Although the *Dsel*^–/–^ mice showed reduced epimerase activity in the brain (89%), kidney (55%), spleen (44%), liver (38%), lung (34%), and skin (24% reduction) compared to that in wild-type tissues, *Dsel*^–/–^ mice exhibited no morphological, histological, or anatomical defects. The level of the IdoUA moiety in CS/DS from the newborn kidney and brain was reduced to 62 and 87% of wild-type mice, respectively, which may be consistent with the finding that *DSEL* was more predominantly expressed in the brain than *DSE* [91,92,117]. DSE may compensate for the loss of functions in DSEL. 

In 2015, Stachtea et al. generated double knockout mice of *Dse* and *Dsel* [118]. They showed complete loss of DS, perinatal lethality, exencephaly, an umbilical hernia, and a kinked tail, suggesting that DS is essential for embryonic development. 

In 2003, Goossens et al. reported that various homozygous or heterozygous mutations in *DSEL*, such as p.Val287Ile, p.Pro673Ser, p.Tyr730Cys, p.Pro942Ser, p.Ile1113Met, and a substitution of adenine to guanine in the 5′-non-coding region 546 bp upstream of the coding region, caused bipolar disorder [92]. In 2011, Shi et al. reported that a single nucleotide polymorphism, 75 kbp upstream of *DSEL*, caused a recurrent early-onset major depressive disorder [119]. Because *DSEL,* but not *DSE*, is predominantly expressed in the brain [91,92,117], the variants of *DSEL* may affect its neuronal functions. 

In 2010, Zayed et al. reported that a congenital diaphragmatic hernia was caused by mutations in *DSEL* [120]. The probands showing diaphragmatic hernia have substitutions of amino acids: p.Met14Ile, p.Asn276Ser, p.Pro683Ser, p.Tyr740Cys, p.Thr842Ser, and p.Asp991Asn. Mouse *Dsel* was expressed in the murine diaphragm muscle on embryonic day (E) 13.5 by in situ hybridization using murine embryo sections [120]. Further analyses may be necessary to understand the pathogenic mechanisms leading to diaphragmatic hernia involving DS as well as DSEL.

### 3.3. CHST14/D4ST1

D4ST catalyzes the transfer of a sulfate group from an active sulfate, PAPS, to the C-4 hydroxy group of GalNAc residues in the repeating disaccharide region of dermatan, [IdoUA-GalNAc]_n_ (Figure 2). In 2001, Tiedemann et al. demonstrated that D4ST activity in human lung fibroblast and bovine articular chondrocyte cultures was higher than that in bovine nasal chondrocyte cultures [87]. Furthermore, Eklund et al. demonstrated that D4ST activity was detected in cell lysate from cultured human embryonic fibroblasts [93]. It has been demonstrated that even though the DSE reaction is freely reversible during the conversion of uronic acids on chondroitin as well as dermatan, neither CS nor DS is recognized by DSE [83]. They proposed that the formation of IdoUA during biosynthesis of DS might be enhanced by 4-*O*-sulfation of the GalNAc residue, thereby being locked in the configuration of IdoUA [87]. 

#### 3.3.1. Identification of *CHST14*/*D4ST1* Gene

In 2000, Yamauchi et al. identified the gene, *carbohydrate sulfotransferase 11* (*CHST11*), encoding chondroitin 4-*O*-sulfotransferase-1 (C4ST-1), which catalyzes the transfer of a sulfate group from PAPS to the C-4 hydroxy group of GalNAc residues in the repeating disaccharide region of chondroitin, [GlcUA-GalNAc]_n_ (Figure 2), based on the protein sequence of purified C4ST from culture medium of rat chondrosarcoma after treatment with trypsin [121]. Subsequently, Hiraoka and Kang et al. identified *CHST12* and *CHST13,* encoding C4ST-2 and C4ST-3, respectively [122,123]. In 2000 and 2001, Baenziger and colleagues identified two genes, *CHST8* and *CHST9*, encoding GalNAc4ST-1 and GalNAc4ST-2, respectively, which transfer a sulfate group from PAPS to a terminal β1,4-linked GalNAc residue in GalNAcβ1-4GlcNAc- on *N*-linked oligosaccharides present on the glycoprotein hormones thyrotropin and lutropin [124,125]. 

In 2001, Evers et al. identified the gene *CHST14,* encoding D4ST-1, through a homology search using the amino acid sequence of the 4-*O*-sulfotransferase family [94]. Mikami et al. demonstrated the substrate specificities of D4ST-1, C4ST-1, and C4ST-2 [95]. Briefly, D4ST-1 generates IdoUA-GalNAc(4-*O*-sulfate)-rich clusters, a typical structure in mature DS chains. C4ST-2 transfers a sulfate to both GalNAc residues in -IdoUA-GalNAc-IdoUA- and -GlcUA-GalNAc-GlcUA- [95]. In 2009, Pacheco et al. demonstrated that the knockdown of D4ST-1 in human lung fibroblasts led to a marked reduction in IdoUA-containing structures in the DS chain [96]. Furthermore, in 2018, Tykesson et al. demonstrated that DSE, but not DSEL, forms a heterocomplex with D4ST-1, which is essential to form longer IdoA-containing chains [126]. Therefore, the cooperation of D4ST-1 and DSE by heterocomplex formation is necessary for the synthesis of repeating disaccharides, [-IdoUA-GalNAc(4-*O*-sulfate)-]_n_, in mature DS.

#### 3.3.2. Knockout Mice of *Chst14/D4st1*

In 2011, Bian et al. generated *D4st1/Chst14*-deficient mice [127]. Neurospheres from *Chst14*^–/–^ mice showed larger diameters and fewer total numbers than those from wild-type mice, caused by dysfunctions in the self-renewal and proliferation of neural stem cells [127]. The epitopes of monoclonal antibody 473HD that recognize the specific sulfation patterns -GlcUA-GalNAc(4-*O*-sulfate)-, -GlcUA(2-*O*-sulfate)-GalNAc(6-*O*-sulfate)-, and -IdoUA-GalNAc(4-*O*-sulfate)- in CS-DS hybrid structures exist on radial glia and neurospheres [128,129]. Furthermore, a decrease in the number of neurospheres formed was observed after chondroitinase ABC or 473HD treatment [130]. In neurospheres generated from *Chst14*^–/–^ mice, distribution of the 473HD epitope was decreased to 58% of that in wild-type neurospheres [127]. These findings indicate that D4ST-1 and DS are important for the self-renewal and differentiation of neural stem cells.

In 2013, Akyüz et al. demonstrated that *Chst14*^–/–^ mice showed a higher proliferation rate and longer cell processes of cultured Schwann cells from nerves and dorsal roots, compared with those of wild-type mice [131]. Although the *Chst14*^–/–^ mice were fertile, one third of the knockout mice died between E16.5 and E18.5 and/or within a few days after birth [131]. The weights of the kidney, liver, heart, tibia, and whole body, but not brain, were reduced in the *Chst14*^–/–^ mice compared with wild-type littermates. These findings suggest that D4ST-1 and/or DS may be essential for early tissue development. 

In 2018, Yoshizawa et al. demonstrated a significantly decreased amount of DS, an abnormal structure of the basement membrane of capillaries, an alteration in the vascular structure with ischemic and/or necrotic-like change, and a reduced weight of the placenta in the placenta derived from *Chst14*^–/–^ mice, compared with wild-types [132]. These findings indicate that D4ST-1 and DS may be indispensable for placental vascular development, and help to elucidate several cases of perinatal lethality caused by a mutation in *CHST14* [133].

In 2021, Hirose et al. demonstrated that DS exhibited a round conformation with wrapped collagen fibrils in wild-type mice, whereas rod-shaped linear CS side chains of decorin observed at one end of collagen fibrils protruded outside the fibrils in *Chst14*^–/–^ mice [134]. Furthermore, the skin tensile strength of *Chst14*^–/–^ mice was lower than that of wild-type mice. These findings indicate that the DS side chain of decorin plays roles in the architecture of collagen fibrils and supporting skin strength.

In 2021, Nitahara-Kasahara et al. demonstrated that elephant teeth, thoracic kyphosis, increased skin fragility, and a kinked tail were observed in *Chst14*^–/–^ mice [135,136]. Moreover, they also found a reduction in DS disaccharide in muscle and myopathy-related phenotypes including spread of the muscle interstitium and variation in fiber size, which may lead to decreased exercise capacity and lower grip strength in *Chst14*^–/–^ mice. These findings suggest that D4ST-1 and/or DS is essential for the myogenesis of skeletal muscle.

#### 3.3.3. Human Genetic Disorders Caused by Mutations in *CHST14/D4ST1*

In 2009, Dünder et al. reported that adducted thumb–clubfoot syndrome in 11 probands from four families exhibited the homozygous mutations p.Val49*, p.Arg213Pro, and p.Tyr293Cys, and the complex heterozygous mutation p.Arg135Gly/p.Leu137Gln in *CHST14* [133]. Adducted thumb–clubfoot syndrome is an autosomal recessive disorder characterized by congenital defects in the heart, kidneys, or intestines, connective tissue fragility with aging, thin and translucent skin, coagulopathy, facial clefting, a typical facial appearance, joint instability, and contracture of the thumbs and feet [137]. Five out of the eleven probands with adducted thumb–clubfoot syndrome died in early infancy or childhood. Furthermore, their studies showed a reduced amount of DS and an increase in CS in fibroblasts from a patient with a mutation of p.Arg213Pro in *CHST14* compared to control subjects [133]. 

In 2005, Kosho et al. reported two probands with generalized joint laxity, scoliosis, fragility, hyperextensibility, readily bruisable skin, atrophic scars, hypotonia, recurrent hematomas, pectus excavatum, and a mild delay of gross motor development [138]. In view of these features, they proposed probands with EDS kyphoscoliosis type VIB. In 2010, Kosho et al. reported six unrelated patients with EDS characterized by progressive multisystem fragility, spinal deformities, recurrent dislocations, progressive skin and joint laxities, multiple congenital contracture of the joints, and characteristic craniofacial features [139]. Subsequently, Miyake et al. identified mutations in *CHST14* encoding D4ST-1 in six probands [140]. The compound heterozygous mutations p.Lys69*/p.Pro281Leu, p.Pro281Leu/p.Cys289Ser, and p.Pro281Leu/p.Tyr293Cys or homozygous mutation p.Pro281Leu were identified in these probands. The recombinant CHST14/D4ST-1 mutants p.Pro281Leu, p.Cys289Ser, and p.Tyr293Cys, which were expressed by COS-7 cells, showed markedly lower D4ST activity than the recombinant wild-type [140]. Consistent with this finding, skin fibroblast cultures from patients with the mutation p.Pro281Leu or p.Pro281Leu/p.Tyr293Cys also showed considerably weaker D4ST activity than healthy subjects [140]. Thus, these substitutions of amino acids affected the enzymatic activity of D4ST-1. Instead of dermatan, a CS chain was produced on the decorin core protein of skin fibroblasts from patients [140]. Because 4-*O*-sulfation in DS inhibits the enzymatic reaction of the conversion of IdoUA into GlcUA by DSE, the lack of 4-*O*-sulfation in DS side chains in probands causes the reverse epimerization reaction, the conversion of IdoUA to GlcUA in dermatan by DSE, resulting in the re-formation of chondroitin, which is sulfated by C4ST to form CS. 

In 2019, Hirose et al. demonstrated the skin pathology of probands with the musculocontractural type of EDS, with both transmission electron microscopy-based cupromeronic blue staining to detect GAG chains and immunostaining of decorin [141]. Collagen fibrils were tightly and regularly assembled in controls; in contrast, they were dispersed in the affected papillary to reticular dermis. Moreover, DS side chains of decorin-PG from control subjects were curved, maintaining close contact with attached collagen fibrils, whereas affected GAGs, CS side chains of decorin-PGs, were linear, stretching from the outer surface of collagen fibrils to adjacent fibrils. These findings suggest that structural alterations of GAG side chains on decorin-PG lead to spatial disorganization of collagen networks. In 2017, Mizumoto et al. demonstrated the absence of urinary DS in patients with pathogenic variants in CHST14/D4ST1 by an anion-exchange HPLC in conjugation with the DS-degrading enzyme chondroitinase B [142]. This method indicates the usefulness of urinary disaccharide compositional analysis of DS chains as non-invasive screening for this disorder.

In 2010, Malfait et al. also independently reported that EDS kyphoscoliotic type VIB in three probands from two families was caused by the homozygous mutations p.Val49* and p.Glu334Glyfs*107 in *CHST14* [143]. They proposed to unify the D4ST1-related syndromes, the novel EDS and the adducted thumb–clubfoot syndrome, as ‘‘musculocontractural EDS’’ [143], which was adopted in the International Classification of Ehlers–Danlos Syndromes in 2017 as EDS musculocontractural type 1 [144,145]. In addition to these mutations in *CHST14*, many probands with EDS musculocontractural type 1 have been reported to date [146,147,148,149,150,151,152,153,154,155,156,157,158,159].

## 4. Conclusions

Recent studies on in vivo functions of DSE, DSEL/DSE2, and CHST14/D4ST-1 using their knockout mice and human cells from patients have partially clarified the pathogenic mechanisms of the disorder as well as the biological significance of DS side chains of DS-PGs. These findings are due to marked contributions by a large number of basic studies since the 1970s by glycobiologists, biochemists, clinicians, and geneticists. Additionally, the knockout mice of these genes help further our understanding of the musculocontractural type of EDS. Advancements in our understanding of the molecular mechanisms involving DS may be necessary for the development of therapeutic approaches and agents.

## Figures and Tables

**Figure 1 genes-14-00509-f001:**
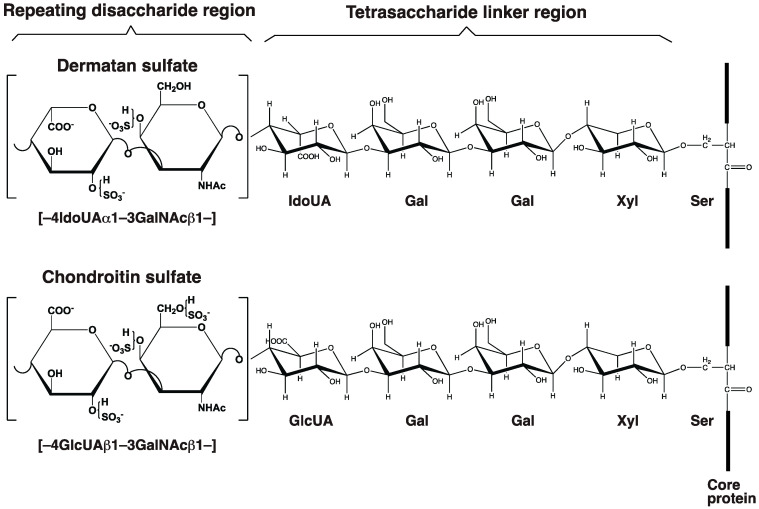
Common GAG-linker region and repeating disaccharide region of DS and CS with their potential sulfation sites. The DS backbone is composed of IdoUA and GalNAc, whereas CS is a stereoisomer of DS that includes GlcUA instead of IdoUA. These carbohydrate residues may be sulfated at various positions indicated by “SO_3_^−^”. It should be noted that sulfation modification of the hydroxy group occurs with various combinations but not all the hydroxy groups. 4-*O*-Sulfation on GalNAc residues in both DS and CS predominantly occurs in mammals. The linkage region tetrasaccharide, IdoUA-Gal-Gal-Xyl-, was isolated from DS-PGs of bovine aorta [25].

**Figure 2 genes-14-00509-f002:**
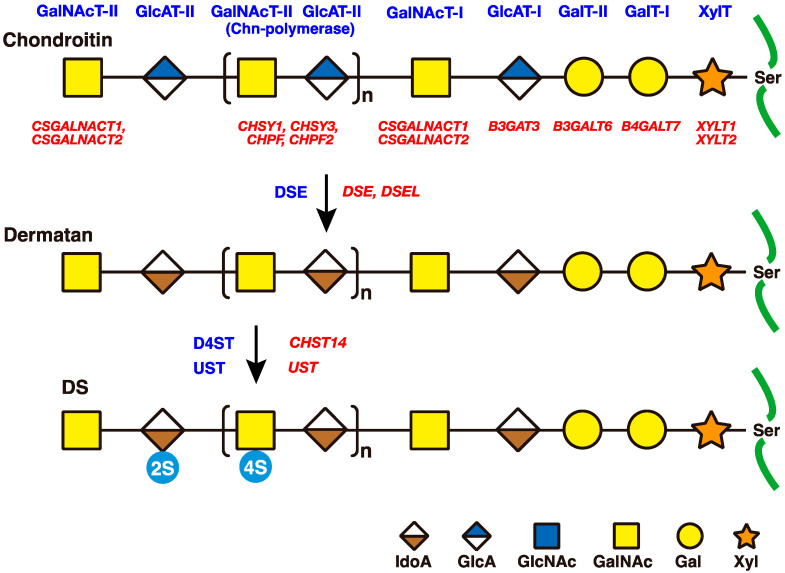
Biosynthetic pathway of DS side chain of PGs by diverse glycosyltransferases, epimerases, and sulfotransferases. Following the biosynthesis of core proteins (green), the common GAG-protein linker region, GlcUAβ1-3Galβ1-3Galβ1-4Xylβ1-, is constructed by XylT, GalT-I, GalT-II, and GlcAT-I on the specific Ser residue(s) of core proteins. After the formation of the linker region, chondroitin synthases build up the chondroitin backbone. Subsequently, the epimerization of GlcUA moieties as well as sulfation of mainly GalNAc and a small proportion of IdoUA residues are catalyzed by DSE as well as D4ST and UST, respectively. Each enzyme and its coding gene are indicated by blue and red characters, respectively.

**Table 1 genes-14-00509-t001:** Representations of core proteins of human DS-PGs.

DS-PGs	Coding Genes	Chromosomal Mapping	Gene ID ^a^	MIM No. ^b^	GAG Type (Number)	Human Genetic Disorders
Decorin	*DCN*	12q21.33	1634	125255 610048	DS/CS (1)	Congenital stromal corneal dystrophy
Biglycan	*BGN*	Xq28	633	300106 300989 301870	DS/CS (2)	Spondyloepimetaphyseal dysplasia, X-linked Meester–Loeys syndrome
Epiphycan	*EPYC*	12q21.33	1833	601657	DS/CS (2–3)	—
Endocan	*ESM1*	5q11.2	11082	601521	DS/CS (1)	—

a, National Center for Biotechnology Information (NCBI); b, Mendelian Inheritance in Man.

**Table 2 genes-14-00509-t002:** General properties of human DSE, DSEL, and CHST14.

Enzyme	Coding Gene	Chromosomal Mapping	Gene ID	MIM No.	Human Genetic Disorders
DSE	*DSE*	6q22.1	29940	605942 615539	Ehlers-Danlos syndrome musculocontractural type 2
DSEL	*DSEL*	18q22.1	92126	611125	Bipolar disorder
D4ST	*CHST14*	15q15.1	113189	601776 608429	Ehlers-Danlos syndrome musculocontractural type 1; adducted thumb–clubfoot syndrome
UST	*UST*	6q25.1	10090	610752	Multiple congenital anomalies of the heart and central nervous system

**Table 3 genes-14-00509-t003:** Pathogenic variants of *DSE*, *DSEL*, and *CHST14*.

Coding Gene	Human Genetic Disorders	Variants
*DSE*	Ehlers-Danlos syndrome musculocontractural type 2	p.Ser268Leu, p.Arg267Gly, p.Tyr320*, p.Val333Cysfs*4, p.Pro384Trpfs*9, p.Tyr867*, and p.Val938Asp
*DSEL*	Bipolar disorder	p.Val287Ile, p.Pro673Ser, p.Tyr730Cys, p.Pro942Ser, and p.Ile1113Met
	Diaphragmatic defect	p.Met14Ile, p.Asn276Ser, p.Pro683Ser, p.Tyr740Cys, p.Thr842Ser, and p.Asp991Asn
*CHST14*	Ehlers-Danlos syndrome musculocontractural type 1; adducted thumb–clubfoot syndrome	p.Arg29Glyfs*113, p.Val49*, p.Lys69*, p.Gln133Argfs*14, p.Arg135Gly, p.Leu137Gln, p.Phe209Ser, p.Arg213Pro, p.Arg218Ser, p.Lys226Alafs*16, p.Arg274Pro, p.Met280Leu, p.Pro281Leu, p.Cys289Ser, p.Tyr293Cys, and p.Glu334Glyfs*107
*UST*	Multiple congenital anomalies of the heart and central nervous system	0.63 Mb deletion in 6q25.1, which includes *TAB2*, *LATS1*, and *UST*

Asterisks indicate the termination codon. The “fs*number” represents a frame shifting change with the new reading flame being open for amino acids and termination codon.

## Data Availability

Not applicable.

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
