# Peer review of "Histories of Dermatan Sulfate Epimerase and Dermatan 4-O-Sulfotransferase from Discovery of Their Enzymes and Genes to Musculocontractural Ehlers-Danlos Syndrome"

_genes, 2023, doi:10.3390/genes14020509_

Round 1
Reviewer 1 Report
This review is an overview of the discovery of the dermatan sulfate and the enzymes involved in its biosynthesis. The authors are knowledgeable and experienced review writers in the field. The review is well organized, clear and accessible for reader outside of the field.
However, some publications relevant to this review are missing.
In the paragraph 3. Roles of DSE and D4ST in biosynthesis of DS, the authors give a summary of the discovery of dermatan sulfate and the DSE, describing the work and contribution of Drs. Malmström and Maccarana’s team. The authors discuss the generation of mice models for both DSE and DSEL enzymes and speculate their relevance during embryonic development. The authors also discuss the possible compensation mechanism between these enzymes in single KO mice. Dr. Maccarana and Dr. Malmstrom have also published data relevant to both aspects in another model system that was not cited by the authors and should be included in this review.
Minor concerns:
Legend Figure 2, the indication of the red and blue annotations is missing.
Author Response
Independent Review Report, Reviewer 1
This review is an overview of the discovery of the dermatan sulfate and the enzymes involved in its biosynthesis. The authors are knowledgeable and experienced review writers in the field. The review is well organized, clear and accessible for reader outside of the field.
However, some publications relevant to this review are missing.
In the paragraph 3. Roles of DSE and D4ST in biosynthesis of DS, the authors give a summary of the discovery of dermatan sulfate and the DSE, describing the work and contribution of Drs. Malmström and Maccarana’s team. The authors discuss the generation of mice models for both DSE and DSEL enzymes and speculate their relevance during embryonic development. The authors also discuss the possible compensation mechanism between these enzymes in single KO mice. Dr. Maccarana and Dr. Malmstrom have also published data relevant to both aspects in another model system that was not cited by the authors and should be included in this review.
Our response:
Thank you for the comment by the reviewer. Another model of DSE and DSEL using knockout mice and Xenopus laevis have been added in the paragraph 3.1 of the manuscript as described below.
“Dse-deficient mice were also showed a reduction in migration of skin-derived dendritic cells from the skin to draining lymph nodes that is a prerequisite for encounter with antigens [105]. Consequently, the initiation of the cellular and humoral immune re-sponse was diminished in Dse–/– mice.”
“In 2016, Gouignard et al., generated Xenopus laevis with knockdown of Dse and Dsel using morpholino antisense oligonucreotides [106]. Although the knockdown of Dse did not affect the formation of early neural crest progenitors, the epimerase activity and extent of neural crest cell migration were decreased in the morphants of Dse, which lead to a decrease in neural crest-derived craniofacial skeleton, melanocytes, and dorsal fin structures. On the other hand, the knockdown of Dsel did not show clear phenotype except for 12% decrease of DSE activity [106].”
Minor concerns:
Legend Figure 2, the indication of the red and blue annotations is missing.
Our response: The following sentence was added in the legend for Fig.2.
“Each enzyme and its coding gene are indicated by blue and red characters, respectively.”
Other collections:
As suggested by Assistant Editor, Dr. Jakub Siudut regarding repetition in our manuscript, the indicated phrases as well as sentences have been changed in the manuscript indicated by red characters.

Reviewer 2 Report
This manuscript by Mizumoto and Yamada provides an extensive historical review of dermatan sulfate (DS) biosynthesis, the function of two key DS enzymes (DSE/DSEL and D4ST1), and the role of pathogenic variants of these enzymes in human congenital disorders. The authors provide insightful perspectives and a useful overview of known genetic mutations responsible for connective disorders and other diseases related to defects in DS biosynthesis. This topic is of high significance and relevance to genetics and biomedicine. Importantly, the manuscript is clearly outlined, well written, and contains appropriate references from the literature. Overall, in this reviewer’s opinion, this review article will meet the interests of the broad readership of Genes. Below, I have included some suggestions to increase readership of the article, including the inclusion of a table providing a list of the known pathogenic variants and connection to human disorders. Please see my reviewer comments below.
1) Please add “the” to the last line of the abstract before “histories”.
2) The authors include an iduronic acid residue in the linker region of DS chains in Figures 1-2. This reviewer is not aware of evidence showing the presence of this residue within the tetrasaccharide linker for CS/DS chains. Major studies in the field (Klein et al MCP 2018; Persson et al MCP 2021) do not include evidence of the presence of IdoA in the linker region. Please provide citation(s) that show evidence of this modification in DS chains. Otherwise, it would be appropriate to replace with GlcA (added by B3GAT3).
3) Table 1: the indication “c” is not necessary as the hyphen demonstrates there are no known human genetic disorders. This reviewer recommends removing this for clarity in the table.
4) There are numerous small paragraphs in section 3.3. This reviewer recommends the authors combine multiple topics into larger paragraphs for clarity.
5) It would be helpful to include an additional table highlighting the known DSE/DSEL and CHST14/D4ST1 variants mentioned in the text and their link to human disorders.
Author Response
Independent Review Report, Reviewer 2
This manuscript by Mizumoto and Yamada provides an extensive historical review of dermatan sulfate (DS) biosynthesis, the function of two key DS enzymes (DSE/DSEL and D4ST1), and the role of pathogenic variants of these enzymes in human congenital disorders. The authors provide insightful perspectives and a useful overview of known genetic mutations responsible for connective disorders and other diseases related to defects in DS biosynthesis. This topic is of high significance and relevance to genetics and biomedicine. Importantly, the manuscript is clearly outlined, well written, and contains appropriate references from the literature. Overall, in this reviewer’s opinion, this review article will meet the interests of the broad readership of Genes. Below, I have included some suggestions to increase readership of the article, including the inclusion of a table providing a list of the known pathogenic variants and connection to human disorders. Please see my reviewer comments below.
1) Please add “the” to the last line of the abstract before “histories”.
Our response: “the” was added in the sentence.
2) The authors include an iduronic acid residue in the linker region of DS chains in Figures 1-2. This reviewer is not aware of evidence showing the presence of this residue within the tetrasaccharide linker for CS/DS chains. Major studies in the field (Klein et al MCP 2018; Persson et al MCP 2021) do not include evidence of the presence of IdoA in the linker region. Please provide citation(s) that show evidence of this modification in DS chains. Otherwise, it would be appropriate to replace with GlcA (added by B3GAT3).
Our response: The IdoA residue of linkage region tetrasaccharide was isolated from dermatan sulfate proteoglycans of bovine aorta in Ref. 25 (Sugahara, et al. Structural studies on the hexasaccharide alditols isolated from the carbohydrate-protein linkage region of dermatan sulfate proteoglycans of bovine aorta: Demonstration of iduronic ac-id-containing components. J. Biol. Chem. 1995, 270, 7204-7212.). As suggested by the reviewer, the following sentence have been added in the legend for Fig. 1.
“The linkage region tetrasaccharide, IdoUA-Gal-Gal-Xyl-, was isolated from DS-PGs of bovine aorta [25].”
3) Table 1: the indication “c” is not necessary as the hyphen demonstrates there are no known human genetic disorders. This reviewer recommends removing this for clarity in the table.
Our response: As suggested by the reviewer, the “c” has been deleted from the Table.
4) There are numerous small paragraphs in section 3.3. This reviewer recommends the authors combine multiple topics into larger paragraphs for clarity.
Our response: As suggested by the reviewer, the subsection 3.2 has been divided into the subsubsections 3.2.1 – 3.2.3 including “Identification of D4ST1 gene”, “Knockout mice of CHST14/D4ST1”, “Human genetic disorders caused by mutations in CHST14/D4ST1”,
5) It would be helpful to include an additional table highlighting the known DSE/DSEL and CHST14/D4ST1 variants mentioned in the text and their link to human disorders.
Our response: As suggested by the reviewer, new Table 3 has been described in the manuscript, which includes DSE/DSEL and CHST14/D4ST1 variants.
Table 3. Pathogenic variants of DSE, DSEL, and CHST14
|
Coding Gene |
Human Genetic Disorders |
Variants |
|
DSE |
Ehlers-Danlos syndrome musculocontractural type 2 |
p.Ser268Leu, p.Arg267Gly, p.Tyr320*, p.Val333Cysfs*4, p.Pro384Trpfs*9, p.Tyr867*, and p.Val938Asp |
|
DSEL |
Bipolar disorder; |
p.Val287Ile, p.Pro673Ser, p.Tyr730Cys, p.Pro942Ser, and p.Ile1113Met |
|
|
Diaphragmatic defect |
p.Met14Ile, p.Asn276Ser, p.Pro683Ser, p.Tyr740Cys, p.Thr842Ser, and p.Asp991Asn |
|
CHST14 |
Ehlers-Danlos syndrome musculocontractural type 1; Adducted thumb-clubfoot syndrome |
p.Arg29Glyfs*113, p.Val49*, p.Lys69*, p. Gln133Argfs*14, p.Arg135Gly, p.Leu137Gln, p.Phe209Ser, p.Arg213Pro, p.Arg218Ser, p.Lys226Alafs*16, p.Arg274Pro, p.Met280Leu, p.Pro281Leu, p.Cys289Ser, p.Tyr293Cys, and p.Glu334Glyfs*107 |
|
UST |
Multiple congenital anomalies of the heart and central nervous system |
0.63 Mb deletion in 6q25.1, which includes TAB2, LATS1, and UST |
Other collections:
As suggested by Assistant Editor, Dr. Jakub Siudut regarding repetition in our manuscript, the indicated phrases as well as sentences have been changed in the manuscript indicated by red characters.

Round 2
Reviewer 1 Report
the authors have satisfactorily addressed the concerns from the reviewer